# A Community Audit of 300 “Drop-Out” Instances in Children Undergoing Ponseti Clubfoot Care in Bangladesh—What Do the Parents Say?

**DOI:** 10.3390/ijerph18030993

**Published:** 2021-01-23

**Authors:** Angela Margaret Evans, Mamun Chowdhury, Sharif Khan

**Affiliations:** 1FFPM RCPS(Glasg), Discipline of Podiatry, School of Science, Health and Engineering, La Trobe University, Bundoora, Melbourne 3083, Australia; 2Walk for Life—Clubfoot Project, Dhaka 1213, Bangladesh; mamun@glencoefoundation.org (M.C.); sikhan@glencoefoundation.org (S.K.)

**Keywords:** Ponseti, relapse, clubfoot, barriers, parents, children

## Abstract

**Introduction:** Drop-out before treatment completion is a vexing problem for all clubfoot clinics. We and others have previously identified better engagement with parents as a crucial method of ameliorating incomplete clubfoot treatment, which increases deformity relapse. **Materials and methods:** The novel use of community facilitators enabled an audit of over 300 families who had dropped-out from a child’s clubfoot treatment. A questionnaire standardized the parent interviews. Parents were encouraged to present for clinical review of their child’s clubfeet. **Results:** When treatment was discontinued for six months, 309 families were audited. A social profile of families was developed, showing that most lived in tin houses with one working family member, indicating low affluence. Family issues, brace difficulty, travel distances, and insufficient understanding of ongoing bracing and follow-up were the main reasons for discontinuing treatment. Overt deformity relapse was found in 9% of children, while half of the children recommenced brace use after review. **Conclusions:** Identifying families at risk of dropping out from clubfoot care enables support to be instigated. Our findings encourage clinicians to empathize with parents of children with clubfoot deformity. The parent load indicator, in parallel with the initial clubfoot severity assessment, may help clinicians to better appreciate the demand that treatment will place on parents, the associated risk of drop-out, and the opportunity to enlist support.

## 1. Introduction

Congenital clubfoot or *talipes equinovarus (CTEV)* is the most significant paediatric orthopaedic deformity [1,2]. Without effective treatment, affected children are disabled for life, impoverished economically and physically, and denied good quality of life [1,3,4]. Given the 250,000 births with CTEV globally each year, with over 80% residing in low-middle-income-countries (LMICs) where 80% receive no treatment, CTEV foot posture is as significant clinically as it is from a public health perspective [1,2,5,6,7].

Drop-out before treatment completion is a well-known and well-cited feature of clubfoot projects globally, and especially in LMICs [1,7,8,9,10,11,12]. The Ponseti method is universally recognized as the “gold standard" treatment for congenital *talipes equinovarus* (CTEV) [6,13,14,15]. The demonstrated results of the Ponseti method are indeed good, however the treatment course is long and includes many clinic visits, which places much demand on young parents and families and has workplace implications [10,16,17]. The Ponseti treatment course generally consists of weekly manual clubfoot correction with cast retention which are repeated approximately six times, involves percutaneous Achilles tenotomy with a three week stability cast, and then maintenance of correction with foot abduction bracing, the basic protocol for which is 23 h/day for three months, followed by “nights and naps” for 3–4 years or until 5 to 6 years of age [18,19,20,21,22,23].

Previous investigations have identified obstacles to treatment completion, including travel distances, costs, time away from work and family, as well as understanding of the overall costs, benefits, and treatment process [10,12,16,17,24]. These same obstacles are also noted by Walk for Life (WFL) in Bangladesh, which as a not for profit project has provided clubfoot care to 27,500 affected children since 2009 [4,25,26].

The concern regarding case drop-out is that incomplete treatment has been shown to increase deformity relapse and related sequelae, which diminish the individual child’s mobility, social assimilation, school and employment access, and quality of life [4,11,27,28].

Recently, we followed-up longer-term drop-out cases and found that problems with initial casting were predictive of drop-out, worse foot posture, and reduced physical function [8]. This study also identified a surprisingly low relapse rate, found disparity regarding typical versus postural clubfoot deformity recognition, and illuminated that many factors for drop-out were due to wider social, family, and treatment-related issues [8].

This subsequent and larger inquiry has addressed the more immediate reasons for drop-out cases from families enrolled at WFL clubfoot clinics in Bangladesh. By introducing a new role of “community facilitator”, WFL both located and evaluated drop-out cases in their homes, and when agreeable to parents, also in WFL clubfoot clinics for clinical review of foot deformity.

Specifically, we aimed to audit the parents’ reasons for discontinuing their child’s clubfoot care at this early stage, to identify pathways by which necessary treatment could be reinstituted, and to examine the children’s feet and gait for deformity correction versus deformity relapse.

The aims of this community audit were:To identify the factors preventing completion of clubfoot treatment at six months of the treatment course;To encourage the parents with children requiring further clubfoot care to access treatment.

## 2. Materials and Methods

This community audit was carried out in 2019 to examine incomplete or “drop-out” cases from WFL clinics across Bangladesh. The audit addressed short term drop-out from cases who missed three appointments or did not attend a WFL clinic for at least six months.

The audit team aimed to identify and locate at least 300 children who had dropped-out. Once participants were identified, the reasons for discontinuing treatment were explored and the parents of children with relapsed clubfoot deformity were encouraged to resume treatment.

Drop-out data were collected from eight of the 32 WFL clinics, including both rural and urban locations. Community facilitators were employed and trained to audit the drop-out children and parents at their homes, using both basic interview and observation approaches.

The audit indicators were:A non-clinical questionnaire; (Appendix A);Bangla clubfoot assessment tool (BCAT), used to evaluate parent satisfaction, assess gait, and for clinical foot examination [4,26,29];The hypothesis underpinning this investigation was that drop-out from treatment can be ameliorated and that factors potentiating drop-out are modifiable. The null hypothesis was that factors inducing drop-out will not be identified or able to be ameliorated to avert drop-out.

### Audit Data Management

All data were entered in spreadsheets (Excel, Microsoft Corp, Redmond, WA, USA), with SPSS version 24 (IBM Corp, Armonk, NY, USA, 2016) used for analysis. Basic descriptive statistics were applied to demographic data, audit assessments, and parent questionnaires. Relationships between variables were analyzed using Spearman’s rho test (Foot Posture Index (FPI) and relapse signs). A comparison between this short-term community audit (*n* = 300) and a previous longer-term study (*n* = 72) in WFL clinics [8] utilized percentage differences.

## 3. Results

In total, five community facilitators audited 311 homes over a four-month period from July to October 2019 and located 309 children who fulfilled the drop-out audit criteria.

The basic demographic data can be found in Table 1. The mean age of the children was 5.2 (1.97) years, with a female/male sex ratio of 101:208, representing the typical male predominance found in clubfoot cohorts [30].

The participating families’ circumstances were explored to provide a living social picture. The number of cohabitating family members was a median value of 4 (range: 2 to 17; from sample of 250 (80.9%)). All of the father’s occupations were recorded (*n* = 309): farmer: 95; labourer: 71; business: 61, driver: 23; service: 17, garment factory worker: 10; other/no work: 35). Most children and families lived in houses constructed from tin sheeting (*n* = 199).

Family types differed, with most being parent–children units (*n* = 225; 72.8%), while others combined generations (*n* = 84; 27.2%). The median number of earning family members was 1 person working, usually the father (range: 0 to 4).

Table 2 compares the drop-out stages and parents’ reasons for treatment incompletion, along with the previous findings from our longer-term study [29]. In both instances, drop-out occurred with brace use (previous study *n* = 72, 100%; current *n* = 95, 30.1%) and associated follow-up appointments (current *n* = 187, 60.5%). It is notable that both reports found family issues and brace difficulty as the main reasons for discontinuing treatment (previous study *n* = 22, 30.6%; current *n* = 86, 27.8%). In the current drop-out audit, the main problem with treatment was the distance (*n* = 54, 17.5%) and frequency of visits (*n* = 28, 9.1%) to the clinics. Further, a lack of understanding (*n* = 23, 7.4%), the impression that the child’s feet were fixed (*n* = 26, 8.4%), and forgetting appointments (*n* = 16, 5.2%) were contributing factors to drop-out. Whilst many parents reported no problems with the treatment, the most reported treatment phase problems were the tenotomy procedure (23.9%) and brace (37.5%) in the current audit cohort. In contrast, the previous study had identified the initial casting phase as that most difficult for parents (and predictive of drop-out [8]).

BCAT was used to identify the children requiring more thorough assessment and further treatment at a WFL clinic. As the audit was undertaken by recently trained Community Facilitator (CF) staff, the BCAT field scores were used only as indicators of relapse and directed further assessment at WFL by trained Ponseti method clinicians.

Parent ratings of clubfoot outcomes are recorded in Table 3. The number of relapsed deformity cases was 25/283 (8.1%). Parent satisfaction was 77.6% (sum of ratings for satisfactory–very good). Further, 94 (30.4%) parents reported that more support was required for financial costs, transport, and further treatment needs, with 38 (12.3%) reporting no further support needs. The clinical reviews directed resumption of treatment in 55.1% (Foot Abduction Brace (FAB) 49.8%, re-cast then FAB 5.3%) of cases. The rate of relapsed clubfoot was only 9% (based on *n* = 283 (91%) of the audit cohort).

## 4. Discussion

We had found previously that parents reported dealing with difficult emotions, and that especially mothers reported sadness, had not told their family of their child’s foot deformity, felt blamed for their child’s deformity, and reported negative reactions from other people (due to their baby having visible casts or braces) [4,8].

Having identified the common causes of distress in parents, the need to better appreciate the parents’ perspective and the factors likely to impede their child’s full clubfoot treatment became very apparent. Specifically, we attempted to better align the parents’ overall load as we undertook the initial clubfoot severity assessment (Figure 1 and Figure 2). Given that parents bring children to the WFL clinics, it is incumbent upon WFL clinicians to forecast and appreciate the many obstacles that often potentiate drop-out from treatment [10,12,16,17,24].

Each component is rated as either 0, 0.5, or 1 (0 normal, 0.5 mildly abnormal, 1 severely abnormal). The six signs are summed to give a total score (maximum = 6), with higher scores indicating worse deformity. Scores 1–4 should reduce with corrective casting, while scores 5–6 usually reduce via Achilles tenotomy.

Each factor is rated and then these ratings are summed as a total score. Early identification of parent load and difficulty enables better understanding and opportunity to provide parents with support to continue and complete their child’s clubfoot correction.

The low relapse rates detected in the children who had dropped-out from treatment in this community audit is perhaps surprising (just 9%; 21% in our previous longer-term study [8]), and reinforces the fact that clubfoot is a non-homogenous condition [31,32,33].

It is fascinating and enlightening to amalgamate the findings from a longer-term drop-out study with the findings of the short-term community audit of drop-out cases from the perspectives of the parents and children. We previously found that problems with corrective casting predicted relapse [8,34] and that parents were distressed if their babies were crying and distressed during this phase, a finding similar to that reported in Uganda [16]. Clinicians may regard casting as somewhat innocuous, as it is the most routine part of the Ponseti corrective treatment course. It is, however, novel for young parents, and a negative experience that is influential and related to drop-out from treatment, as well as increased likelihood of deformity relapse. Therefore, crying during the initial casting phase is to be avoided, and generally can be if clinicians take time to settle the parents and child and consider their comfort (e.g., set up casting needs prior to arrival, parent seated and holding baby, allow feeding, use gentle manual handling) [26].

From this community audit, our previous study [8], and other reports [16,35,36,37], it is clear that family factors loom large and that phone connectivity is often fraught [4].

The emerging role of community facilitators in reducing drop-out once the bracing phase starts, when most adherence falters [38,39,40,41,42], means additional staff and associated costs, but potentially may also in better rates of treatment completion, equating to less resource wastage and better results for more children in correcting clubfoot deformity, allowing them to lead happy and productive lives.

Drop-out from the clubfoot treatment course needs to be addressed, and awareness of parent’s feelings (fear, guilt, shame, etc.) may be beneficial [8,10,16,17,43,44,45,46]. Extra support via parent groups and group review sessions during the brace phase are envisaged by WFL, as well as the incorporation of parents whose children have completed clubfoot correction to provide credible experience and support. Social media groups are another avenue for support, and the WFL Facebook family page is well utilized.

It is curious as to why do some drop-out cases relapse and others do not [11,28,37,47]. Whilst it is encouraging in both this short-term audit and the previous longer-term review that the relapse rate following incomplete treatment was much lower than anticipated, it also raises questions of treatment “dosage”, especially in the repeatedly fraught bracing phase [48,49,50].

Can we identify antecedent factors that dictate less or more brace need? It has been reported that the mean duration of treatment with the brace to achieve non-relapse of clubfoot deformity and to avoid surgery was 33 months [21]. The randomized Clubfoot Foot Abduction Brace Length of Treatment Study, comparing the effectiveness of two versus four-year foot abduction bracing, may provide valuable guidance [51].

However, perhaps the need for simultaneous and more functionally based adjunctive intervention could help with the well-reported problems with brace tolerance and resulting drop-out [19,52,53]. The WFL clinics now direct physical therapy in the form of simple manual exercises (ankle dorsiflexion, abduction, eversion) as adjunctive treatment, and similarly encourage a full squat position for play and eating times in older children.

Another aspect of the complete Ponseti method is provision of a tibialis anterior tendon transfer (TATT) procedure, which if availed prior to functional gait relapse (heel inversion, forefoot adduction) can avert deformity recurrence [54,55]. WFL clinics in Bangladesh have not had adequate access with this surgical technique. It is worth noting that the TATT procedure is technically particular, whilst a less invasive option used by some surgeons is the easier peroneus longus–peroneus brevis augmentation procedure [56]. This is an avenue of development for the WFL project, where the scale of need and applicable surgical access are challenging.

The initial severity rating for clubfoot deformity is widely practiced and provides a clinical roadmap as treatment progresses, with a decreasing score indicative of improvement of the physical deformity (Figure 1) [57,58]. Given the findings of this community audit, our previous investigations, and those of other authors, we have now developed a parent load indicator for use alongside the initial clubfoot severity score, which will help clinicians to assess the demand that treatment will place on parents and the associated risk of drop-out (Figure 2). By identifying parents at risk, supports can be put in place, including counselling, travel cost supplements, communication with other parents in group review sessions, and social media contacts.

Future investigation will address the effects of adjunctive approaches and the effects of identifying and better supporting parents from the outset, so as to reduce drop-out rates and achieve the best results for the most children.

There were some limitations to this community audit, which was based upon case convenience and ease of travel access for the CF, but did cover eight districts across Bangladesh. Given the nature of this audit, the assessment questionnaire was developed pragmatically in response to need and available CF staff. Similarly, the BCAT was used by non-clinical personnel, and hence only as a directive tool, which precluded accurate scores being derived. Further, the CF staff in most cases questioned both parents, yet in some instances only one parent was available. It is not known whether this affected the arising data.

## 5. Conclusions

We need to better understand and work with parents more closely from the very first consultation in clubfoot settings. Relapse has been predicted by problems with initial casting, especially when the child is distressed and cries a lot, worrying and upsetting young parents.

Given that parents are already feeling a range of emotions (guilt, shame, blame, fear, etc.), all clinical staff need to focus equally on the parents’ concerns and work and family demands, as they do on the initial severity of the child’s clubfoot deformity. The newly developed parent load indicator is now being evaluated by WFL, as are appointment reminders, parent support groups, cost-sharing models, and staff updates.

Thus, the hypothesis that drop-out from treatment can be ameliorated and that factors potentiating drop-out are modifiable has been in part supported, with the null hypothesis not being supported.

The need to work with, support, and maintain communication with young parents is paramount. This may alter clinic staffing to include support staff for parents and widen the family focus of clubfoot clinics more generally. This will be challenging in the already busy LMIC clubfoot clinics, however will potentially waste less resources and may achieve better results for more children.

## Figures and Tables

**Figure 1 ijerph-18-00993-f001:**
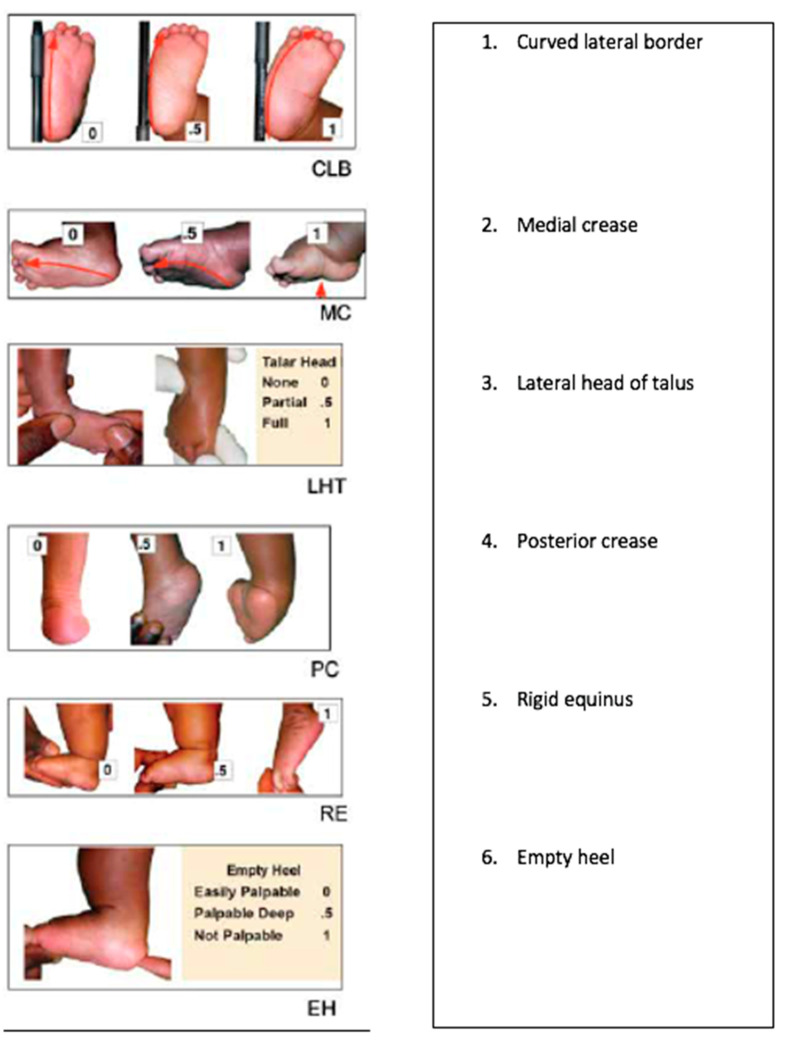
Initial clubfoot severity is rated using six clinical signs of foot deformity (Pirani score).

**Figure 2 ijerph-18-00993-f002:**
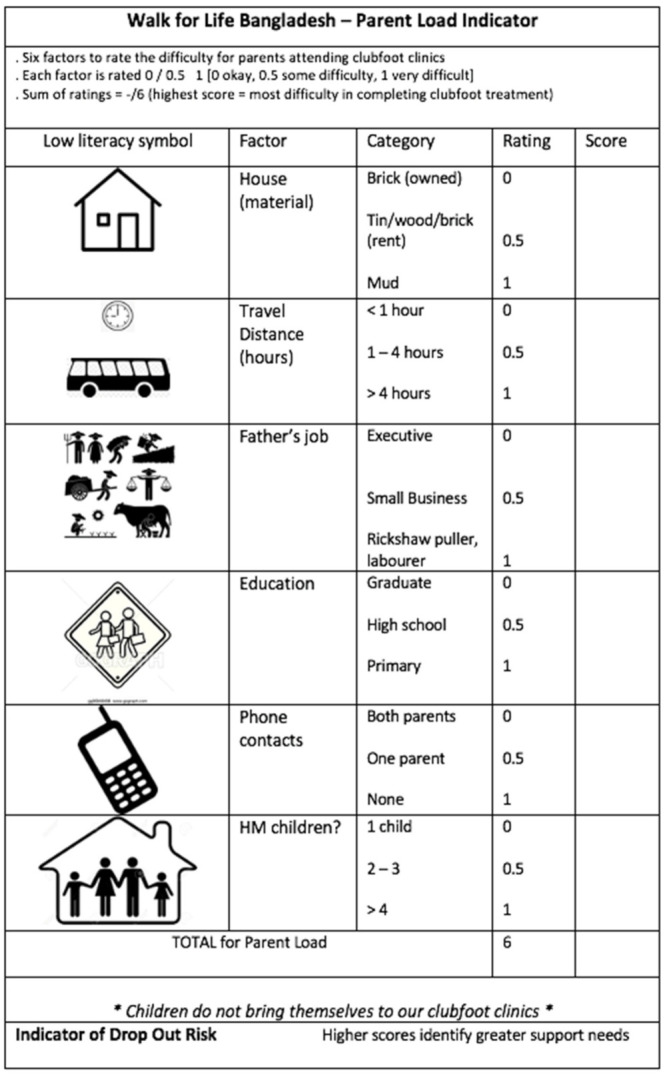
Initial parent load can be rated simultaneously to initial clubfoot severity score.

**Table 1 ijerph-18-00993-t001:** Demographic data from the community audit.

Sample	309
Time since drop-out	6 months
Male Female	208 (67.3%) 101 (32.7%)
Age (years) Mean Range	5.2 (SD 1.97) 0.7–11
Housing materials Brick Tin Wood Mud	74 199 22 18

**Table 2 ijerph-18-00993-t002:** Treatment stage and parents’ reasons for drop-out.

	Previous Study	Current Study	% Difference Current vs. Previous Samples
Drop-out Sample	72	309	
Treatment stage of drop-out Casts PAT/casts Brace Follow-up	0 0 72 (100%) 0	17 (5.5%) 10 (3.2%) 95 (30.7%) 187 (60.5%)	−5.5% −3.2% 69% −60.5%
Reasons for drop-out Family issues Finances Difficulty with brace Problems with treatment Long term, many visits Bad staff Fear of tenotomy Cannot find clinic Child ill Too far to travel Lack understanding child seemed ok forgot appointment	22 (30.6%) 5 (69.4%) 30 (41.6%) 11	86 (27.8%) 22 (7.1%) 24 (7.7%) 28 3 2 6 14 54 23 26 16	−2.7% −30.3% −71.4%
Treatment phase found difficult None Casts PAT/casts Brace	50 (69.4%) 10 (13.8%) 2 (2.7%) 5 (6.9%)	68 (22.0%) 43 (13.9%) 74 (23.9%) 116 (37.5%)	47.4% −0.1% −21.2% −30.6%

PAT = Percutaneous Achilles Tenotomy. The brace and follow-up periods are when drop-out occurs, as indicated by both short-term and longer-term reviews. Family factors appear to be the main obstacles to continued clubfoot care. Cited treatment problems include focus of travel demands, a lack of understanding, forgetting appointments, and child illness. Underlined text denotes a treatment stage, or family reason.

**Table 3 ijerph-18-00993-t003:** Parents’ ratings of clubfoot outcomes.

Functional Level of Children (*n* = 309)					
Test	Single leg standing		Squatting	Weakness	Pain
Yes	201		179	107	66
No	108		130	202	243
Parent ratings (*n* = 309)		Support needed		Type of support needed	
Not satisfactory	69	Yes	94	Financial	48
Satisfactory	32	No	215	Repeat Treatment	12
Average	42			Transport	35
Good	95			None	38
Very Good	71				
Clinic reviews, *n* = 283 (91% sample) Resume FAB Re-cast, then resume FAB OK, discharged Relapse deformity		49.8% 5.3% 27.9% 25/283 (9%)			

The community audit of children who had dropped-out from completing clubfoot treatment identified more functional deficits than pain (21.3%). Dissatisfaction with treatment was reported by 69/309 (22.3%), with support needs stated by 30.4%. The main support requests were for travel and financial assistance. Clinical review saw instigation of bracing in half of the children, with a further quarter needing no further treatment and being discharged. Deformity relapse was identified in 9% of early drop-out cases.

## Data Availability

No additional data available.

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
