# Peer review of "A Community Audit of 300 “Drop-Out” Instances in Children Undergoing Ponseti Clubfoot Care in Bangladesh—What Do the Parents Say?"

_ijerph, 2021, doi:10.3390/ijerph18030993_

Round 1

Reviewer 1 Report

Dear Authors,

I think you have made many good improvements to your manuscript. Without a doubt, it has improved the quality of a very interesting article for the scientific community. Congratulations.

Reviewer 2 Report

Thank you for the revision work. I don't have further comments. 

This manuscript is a resubmission of an earlier submission. The following is a list of the peer review reports and author responses from that submission.

Round 1

Reviewer 1 Report

This study interviewed parents of children with clubfoot who discontinued treatment in Bangladesh. The study found that reasons for dropout included long distance to the clinic, frequency of care, and misunderstanding of the treatment progress, etc. These findings provide insights for clinicians to target patients with higher risk of treatment discontinuation. I have a few comments/questions:

  1. Do you consider the indications from this study are applicable internationally, or mainly to Bangladesh?
  2. The results presented do not communicate very well. Some essential data was shown in supplementary while others that did not seem to be study findings were placed in format text (e.g., Figure 1). Figure 1 and Table 5 were not mentioned in results (and did not seem to be study findings). Suggest to re-organize the result presentation so that they communicate standing alone.
  3. What is the study setting (e.g., geographic coverage)? Is this a nationally representative study sample?
  4. Did you interview both parents of the patient? If both, how do you reconcile the answers?
  5. Your presented results from a previous study, however, no background information was available in the current manuscript. Why did you compare the two studies? How do you interpret the differences?

Reviewer 2 Report

This article provides interest to understand the role of parents as a crucial factor in the abandonment in the treatment of clubfoot.
It is an interesting topic that serves to understand the nature and scope of this problem in Bangladesh. The article title and abstract are adequate. However, the manuscript should establish research hypotheses according to the established objectives and include them in the methodology section. At the end of the work, in the conclusions, the authors must indicate the degree of compliance or not with these hypotheses. Likewise, it is recommended that the authors complete their work with a linear regression analysis that makes it possible to emphasize the statistical relationship between the predictor and response variables.

On the other hand, the document is successful in presenting the state of the matter, attending to primary international sources, although it is recommended to increase the number of citations of works published in the last 3 years.

Furthermore, this work carries out a study with an adequate sample (n = 309).